# LSR-YOLO: A lightweight and fast model for retail products detection

**Yawen Zhao[1], Mahmud Iwan Solihin[1]\*, Defu Yang[2], Bingyu Cai[1,3], Li Sze Chow[1], Dini Handayani[4], Anton Satria Prabuwono[5]**

1 Faculty of Engineering, Technology and Built Environment, UCSI University, Kuala Lumpur, Malaysia, 2 College of UAV, Guangzhou Civil Aviation College, Guangzhou, Guangdong, China, 3 School of Advanced Manufacturing, Shantou Polytechnic, Shantou, China, 4 Faculty of ICT, International Islamic University Malaysia, Gombak, Selangor, Malaysia, 5 Department of Computing, Faculty of Science, Management & Computing, Universiti Teknologi PETRONAS, Seri Iskandar, Perak, Malaysia

\* mahmudis@ucsiuniversity.edu.my

## Abstract

Advanced computer vision techniques, particularly deep learning–based object detection, are enhancing the accuracy and efficiency of product identification in retail settings, driving the integration of intelligent systems within urban environments and smart cities. To address the high computational cost and slow detection speed of existing methods, this study proposes LSR-YOLO, a lightweight object detection framework based on YOLOv8n, designed for deployment in robots and intelligent devices. The model introduces architectural optimizations, including the CSPHet-CBAM attention module, to strengthen feature representation, followed by a channel pruning algorithm tailored to the new architecture to reduce redundancy while maintaining accuracy. Experiments on the Locount dataset demonstrate that LSR-YOLO achieves an inference speed of 357.1 FPS with mAP50 of 72.2% and mAP50-95 of 47.8%. Compared with the baseline YOLOv8n, LSR-YOLO increases inference speed by 246.7 FPS, making it substantially faster and more suitable for real-time retail applications. With only 2,114,768 parameters and 6.6 GFLOPs, it is also significantly lighter than advanced models such as YOLOv11. Furthermore, validation on the COCO dataset confirms the model's superior generalization ability, underscoring its advantages in both accuracy and computational efficiency.

## 1. Introduction

In retail scenarios, product recognition is one of the key technologies for achieving intelligent retail [1]. As the demand for automation and intelligence in the retail industry continues to increase, the application of intelligent robots in areas such as product recognition and customer service [2,3] has gradually become a research hotspot. With the continuous evolution of the retail industry, especially the integration of e-commerce and offline retail, the variety of products and display methods

**Data availability statement:** https://isrc.iscas.ac.cn/gitlab/research/locount-dataset, https://cocodataset.org/#home.

**Funding:** The author(s) received no specific funding for this work.

**Competing interests:** The authors have declared that no competing interests exist.

in retail scenarios have become increasingly diversified. Traditional product recognition methods are no longer able to meet the demands of modern retail. Intelligent robots, by automatically recognizing products and updating inventory in real-time, can significantly improve operational efficiency and reduce artificial errors [4]. The widespread adoption of self-checkout systems requires product recognition technologies to efficiently and accurately identify various types of products and perform rapid transactions.

Traditional product recognition methods primarily include techniques such as template matching [5] and machine vision [6]. While these methods have achieved some success in specific scenarios, they exhibit significant limitations when dealing with the complexity of retail environments. Template Matching-Based Methods recognize objects by comparing the input image with predefined templates [7]. However, template matching is extremely sensitive to factors such as object rotation, scaling, and lighting variations. On the other hand, machine vision methods capture product images and process them using methods like edge detection, morphological processing, and feature point matching for product recognition [8]. While these methods can handle simpler tasks, they often struggle to cope with the complexity of products and environmental changes in complex retail scenarios.

In recent years, deep learning has emerged as a transformative technology for addressing challenges in retail product recognition and various applications [9–12]. Particularly, Convolutional Neural Networks (CNNs) [13] have become one of the most powerful tools for product identification and classification tasks in dynamic retail environments. Numerous studies have demonstrated the superior performance of CNNs in object detection, classification, and segmentation tasks, positioning them as the foundation for modern retail product recognition systems. One of the key advantages of deep learning, particularly CNNs, is their capacity for end-to-end learning and processing. This enables the model to automatically extract and learn intricate product features, thus exhibiting robust generalization abilities when confronted with complex retail conditions such as diverse lighting, product occlusions, and variations in product appearance [14]. Recent advancements in deep learning have further enhanced its application to retail environments. For instance, YOLO (You Only Look Once) [15] and SSD (Single Shot Multibox Detector) [16] are real-time object detection models that significantly improve retail product recognition efficiency. They simultaneously localize, classify, and count products in an image, making them ideal for fast-paced retail environments where quick decision-making is crucial. Moreover, techniques like data augmentation and transfer learning have been successfully employed to address the inherent challenges of retail product recognition, such as diverse product variations and environmental factors. These strategies allow models to generalize across a wide range of products and adapt to the variability in retail settings [17]. In practical implementations, CNN-based models have shown remarkable accuracy in retail product recognition, even when faced with complex visual conditions. For example, recent research has demonstrated how CNNs can be trained on large-scale datasets to achieve high precision in recognizing products in real-time, as seen in the application of models such as YOLOv4 [18]

and EfficientDet [19]. These models have been optimized for both accuracy and speed, making them highly effective in real-world retail scenarios.

Although deep learning models have made significant progress in accuracy, many high-precision models incur substantial computational overhead, especially when dealing with large-scale product data [20]. In such cases, real-time performance and computational efficiency may become bottlenecks. In practical retail systems, product recognition needs to be completed within a few milliseconds to a few seconds, making the improvement of recognition efficiency crucial. To address these issues, this study is driven by the engineering application requirements of shopping robots and proposes a lightweight object detection method for merchandise, based on YOLOv8n and suitable for deployment on edge computing devices. The method utilizes a channel pruning algorithm for model optimization, ensuring high accuracy while enhancing real-time detection performance. The contributions of this article are followed as:

(1) Using DWConv as convolution layers with different scales in Neck network separates the convolution operations on the spatial and channel dimensions, significantly reducing the computational cost;

(2) A attention mechanism called the CSPHet-CBAM is introduced to prevent significant accuracy degradation that may occur after the Neck is lightweighted;

(3) To simplify the bulky backbone network, ADown is introduced for downsampling optimization using average pooling and max pooling, and both the number of parameters and computational complexity are significantly reduced;

(4) After training the YOLOv8n-improved model for retail product defect detection, model compression is applied, making the model lighter while maintaining model accuracy.

## 2. Methods

The research roadmap for retail product object detection models is illustrated in Fig 1. First, raw images are randomly selected from the Locount dataset [21] and preprocessed to enhance data quality and ensure compatibility with subsequent model training. Next, to improve the detection accuracy of the YOLOv8n model while reducing computational costs, we introduce depthwise separable convolution (DWConv), an improved CBAM attention mechanism (CSPHet-CBAM), and a dimensionality reduction operation (ADown) to optimize the architecture and feature extraction capabilities of YOLOv8n. Subsequently, to enhance the model's lightweight characteristics and detection accuracy, the YOLOv8n-improved model undergoes sparse training, channel pruning, and fine-tuning, leading to model compression, increased inference speed, and the development of LSR-YOLO. Finally, comprehensive ablation and comparative experiments are conducted using both the YOLOv8n-improved and the compressed LSR-YOLO on the preprocessed Locount dataset. Additionally, the generalization ability of the models is evaluated on the large-scale COCO dataset [22] to validate their adaptability in complex retail environments.

### 2.1 Lightweight improvement model based on YOLOv8n

YOLOv8 is an advanced version of the You Only Look Once (YOLO) object detection series. Building on the strengths of previous generations, it incorporates several innovative improvements, leading to significant enhancements in both performance and efficiency. Compared to two-stage algorithms such as R-CNN [23], Faster R-CNN [24], and Mask R-CNN [25], YOLOv8 offers superior real-time performance and faster detection speed. The YOLOv8 architecture is divided into YOLOv8n [15], YOLOv8s [26], and YOLOv8l [27]. Among these, YOLOv8n is most suitable for applications that require high real-time performance but have lower accuracy demands. With the least computational complexity and parameter count, and the fastest inference speed, YOLOv8n is selected for this study.

The YOLOv8n network architecture primarily consists of the input layer, backbone network, neck network, and head network. However, for the shopping robot system, real-time performance is crucial during product detection and

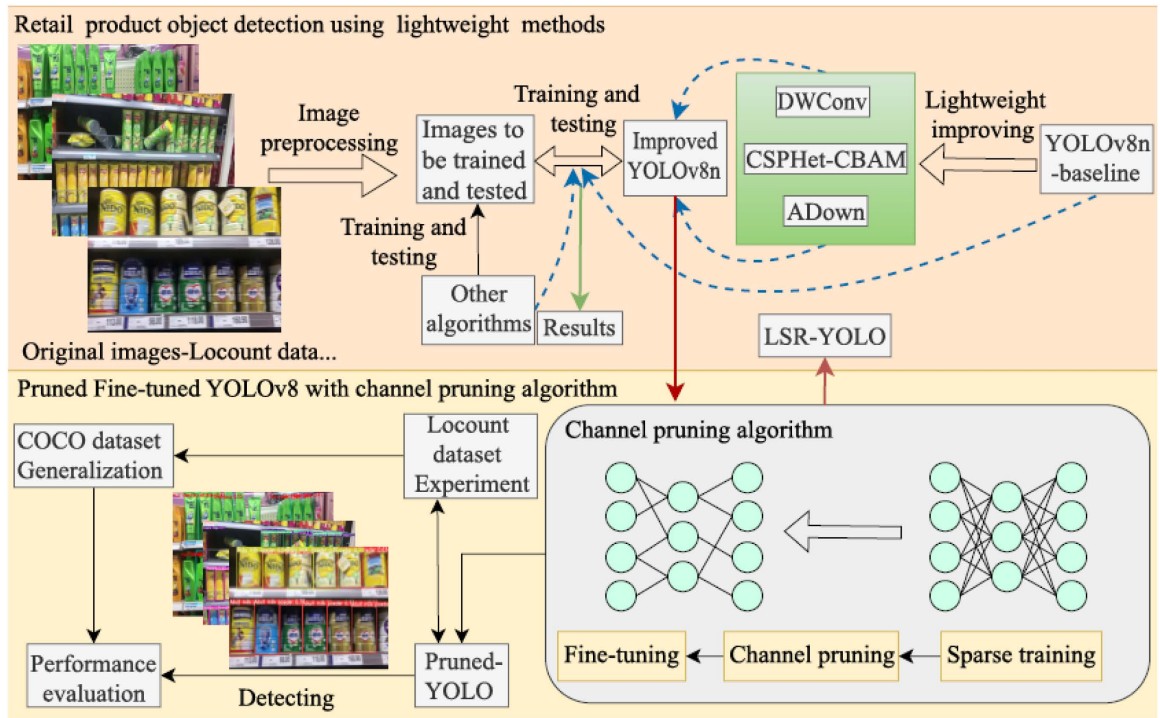

**Fig 1. Roadmap for retail product object detection using lightweight methods.**

recognition. Therefore, lightweight optimization is a key consideration in the model design. The proposed improvements include replacing the convolutional module (Conv) in the neck with a concise DWConv module and integrating the CSPHet-CBAM attention mechanism into the C2f module of the neck. Additionally, an ADown module is introduced in the backbone to replace certain Conv, aiming to reduce the model's complexity. The detailed design and implementation of these lightweight strategies will be thoroughly discussed in subsequent sections. The overall structure of the final improved YOLOv8n model is illustrated in Fig 2.

**2.1.1 DWConv.** DWConv [28] is a depthwise convolution operation that applies a separate convolution kernel to each input channel independently, rather than using a shared convolution kernel across all input channels, as is shown in Fig 3. When performing convolution on the input feature map, each convolution kernel in DWConv processes only a single input channel, instead of processing all channels simultaneously as in traditional convolutions. This means that for each channel, the convolution kernel performs local receptive field operations within that channel, thereby reducing the computational complexity.

In standard convolution, the size of the convolution kernel is ($C_{in}$, $C_{out}$, $H$, $W$), where $C_{in}$ is the number of input channels, $C_{out}$ is the number of output channels. In DWConv, the number of convolution kernels equals the number of input channels $C_{in}$, with each kernel performing convolution only once per channel, significantly reducing the computation. For each input channel, the size of the convolution kernel is (1, 1), which further reduces the computational load [29]. In the end, a batch normalization layer is added to standardize the convolution results, eliminating potential biases and enhancing training stability. Subsequently, a nonlinear transformation is applied to the output feature map through an activation function. The SiLU (Sigmoid Linear Unit) [30] activation function exhibits smooth nonlinear characteristics, enhancing the model's ability to capture complex features.

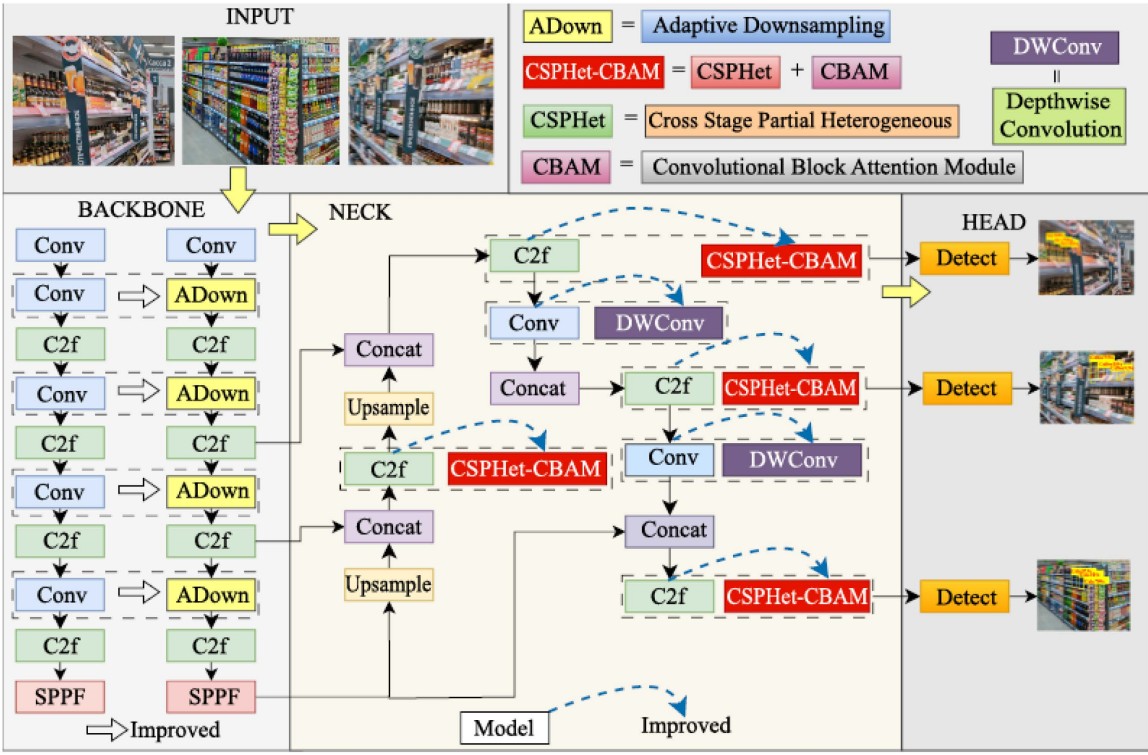

**Fig 2. Overall structure diagram of YOLOv8n-improved model for the retail product object detection.**

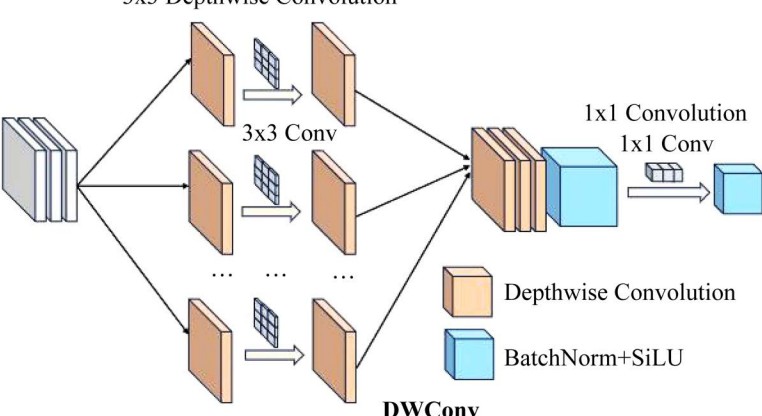

**Fig 3. Structure diagram of DWConv.**

In this study, DWConv is integrated into the Neck structure in YOLOv8n, replacing traditional and dense convolution operations. The DWConv is inherited from the standard convolution layer class and determines the groups of each convolution kernel through GCD (Greatest Common Divisor) calculation [31], thereby implementing depthwise convolution operations. Using DWConv allows the separation of convolution operations in both spatial and channel dimensions, significantly reducing computational complexity. By employing DWConv with different kernel sizes, it is possible to extract

more diverse features across varying spatial scales, while also capturing information from different scales. This enables the module to execute more efficiently and reduce the computational burden, ultimately achieving reduced computational and memory overhead. In scenarios with numerous object categories and complex backgrounds, the DWConv module efficiently extracts key features of products, ensuring high accuracy in object detection.

**2.1.2 CSPHet-CBAM.** To avoid the significant accuracy degradation potentially caused by shallow DWConv networks in the Neck, we designed the CSPHet-CBAM, a novel attention mechanism. CSPHet-CBAM is a novel module that combines CSPHet [32] and CBAM [33], designed to enhance the model's capability in processing complex visual information, particularly in product recognition tasks.

The CBAM module enhances feature extraction by merging two attention mechanisms: Channel Attention and Spatial Attention. These mechanisms enable the model to focus more effectively on key regions, particularly when identifying important items and local features, as shown in Fig 4. The Channel Attention adaptively assigns different weights to each channel, emphasizing the most critical features. In retail environments, where certain products may exhibit distinctive features in specific channels, the Channel Attention helps the model prioritize these significant details.

The Spatial Attention, on the other hand, generates a spatial attention map that reflects the spatial correlations among the features [34]. This allows the model to focus on critical regions of an object, especially in challenging scenarios where the object is partially occluded or surrounded by complex backgrounds. By combining these two attention mechanisms,

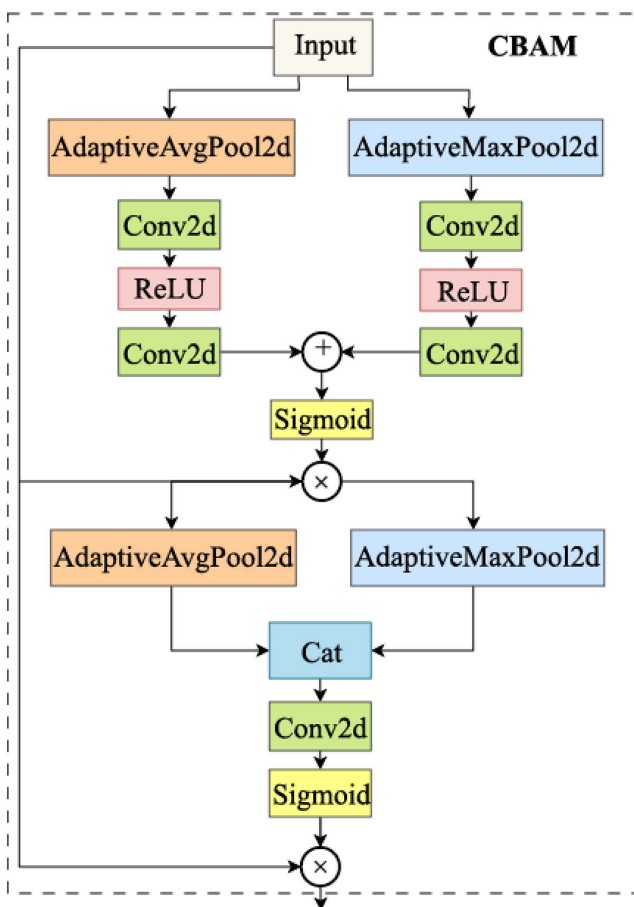

**Fig 4. Network structure of CBAM module.**

CBAM improves the model's ability to selectively highlight and weight input features, enhancing its sensitivity to fine product details and making it more effective in detecting key items in cluttered environments.

In addition, Fig 5 introduces HetConv [35], an innovative convolutional structure that combines 1x1 and 3x3 convolution operations to effectively handle varying feature scales. Specifically, the 1x1 convolutions serve to reduce the number of channels, thereby decreasing the dimensionality and computational load. Meanwhile, the 3x3 convolutions capture local features and preserve spatial information, enabling the model to maintain a rich feature representation. By incorporating these two types of convolutions, HetConv strikes an effective balance between preserving important feature information and reducing computational complexity, which is particularly beneficial when processing high-dimensional data. This hybrid convolution approach not only retains essential details but also improves computational efficiency, making it well-suited for real-time applications such as retail product detection.

By replacing the C2f module, CSPHet-CBAM has the HetConv (Heterogeneous Convolution) structure, which combines multiple convolution operations of different sizes. And it provides a highly flexible and efficient solution to capture multi-scale features. This hybrid convolution approach is distinct from previous works that use fixed or uniform convolution operations, as it dynamically adapts to varying feature scales, reducing computational complexity without sacrificing accuracy, particularly when handling high-dimensional data in settings with many channels.

Furthermore, the integration of CBAM within the CSPHet-CBAM framework introduces a dual attention mechanism (channel and spatial attention) that is designed not just for general feature enhancement, but specifically for distinguishing critical object features in cluttered environments. Unlike previous methods for retail products that apply attention mechanisms independently, CSPHet-CBAM strategically combines both attention types to improve the model's focus on essential details, especially under challenging conditions like partial occlusions or backgrounds with high noise. The channel attention mechanism allows the model to dynamically emphasize the most relevant feature channels, while the spatial attention mechanism enables the model to concentrate on significant spatial regions, boosting its performance in object detection tasks with intricate backgrounds.

The Cross-Stage Partial (CSP) design further distinguishes CSPHet-CBAM by incorporating partial residual connections that facilitate more efficient feature flow across stages. This design mitigates the gradient vanishing problem by

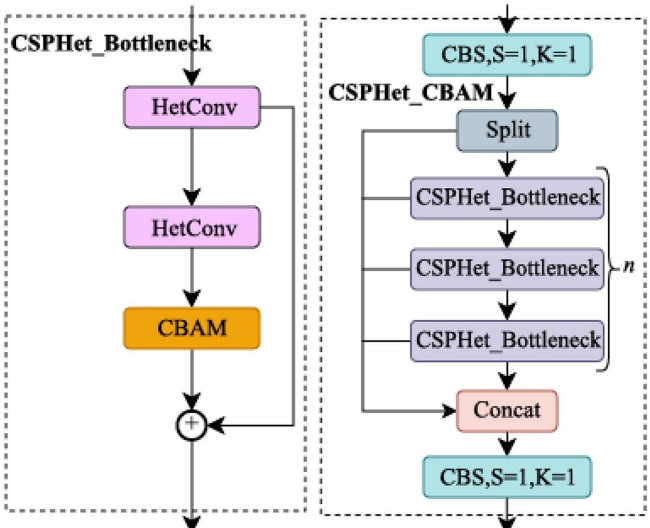

**Fig 5. Structure diagram of CSPHet bottleneck and CSPHet-CBAM.**

ensuring smoother gradient propagation, improving model convergence, and increasing robustness, especially in deep architectures.

**2.1.3 ADown in backbone.** The core design concept of ADown [36] is to compress the input features through downsampling operations and combine them with convolution to extract effective features. ADown splits the input feature map along the channel dimension into two parts, which are processed separately. This operation effectively reduces the computational load and optimizes feature extraction efficiency. As is shown in Fig 6, the first part applies a 3x3 convolution followed by a stride-2 downsampling operation [37] (AvgPool2d) to reduce the feature size. The second part first performs 3x3 max pooling [38] (MaxPool2d) for spatial dimensionality reduction, and then further compresses the channel information through a 1x1 convolution. Finally, ADown concatenates the processed feature maps from the two parts, preserving the original information and enhancing feature representation. The number of parameters and computational complexity of the ADown module are primarily determined by the 3x3 convolution layer and the 1x1 convolution layer. Since the channel dimension of the feature map is divided before downsampling, both the number of parameters and computational complexity are significantly reduced.

In addition to its lightweight design, ADown employs a hybrid pooling strategy that integrates average pooling and max pooling. This combination is strategically used to capture a wider range of features from the input data. Average pooling, which typically focuses on smoothing and capturing background information, helps retain broader contextual features. On the other hand, max pooling focuses on identifying the most significant elements, excelling at preserving fine-grained texture and local features. By using both pooling techniques in tandem, ADown effectively enhances the model's ability to extract a more comprehensive and varied set of features, which improves the representation of the product data.

Moreover, ADown diverges from traditional convolutional architectures by using a parallel downsampling operation rather than a serial structure. This parallel approach ensures that more critical information is retained across layers while reducing spatial dimensions. This design choice improves computational efficiency and boosts the model's performance, particularly in real-time tasks like retail product detection.

Let the size of the feature map and the downsampled feature map be denoted as $H \times W \times C$, where $H$ represents the height, $W$ denotes the width, and $C$ denotes the number of channels of the feature map, $C_{in}$ denotes input channel number, $C_{out}$ denotes output channel number. The number of parameters $P_{Conv}$ and computational complexity $F_{Conv}$ of the downsampling 3x3 convolution layer with a stride of 2 can be mathematically expressed as follows:

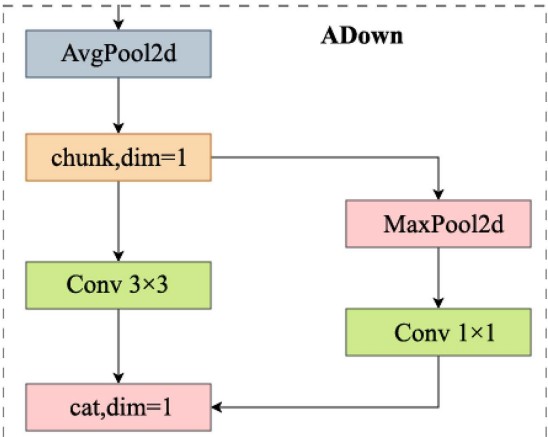

**Fig 6. Structure diagram of ADown.** It is used in YOLOv8n backbone to replace Conv.

$$P_{conv} = (3 \times 3 \times C_{in} + 1) \times C_{out} \tag{1}$$

$$F_{conv} = (\frac{H}{2} \times \frac{W}{2}) \times C_{in} \times C_{out} \times 9 \tag{2}$$

Then, the number of parameters $P_{ADown}$, computational complexity $F_{ADown}$, can be mathematically expressed as follows:

$$P_{ADown} = (9 \times C_{in} + 1) \times C_{out} + C_{in} \times C_{out} \tag{3}$$

$$F_{ADown} = (H \times W \times C_{in} \times 9 \times C_{out}) + (\frac{H}{2} \times \frac{W}{2} \times C_{out}) + (\frac{H}{2} \times \frac{W}{2} \times C_{in}) + (\frac{H}{2} \times \frac{W}{2} \times C_{in} \times C_{out}) \tag{4}$$

By comparing equations (1) and (3), it can be observed that the ADown module introduces an additional 1x1 convolutional layer, but due to the splitting of the input feature map into two parts for processing, the overall number of parameters remains relatively low. Similarly, comparing equations (2) and (4) reveals that the ADown module reduces computational complexity by processing the feature map in two stages and employing pooling layers. Therefore, although the computational complexity of the ADown module is relatively higher, it offers improved feature extraction and computational efficiency compared to traditional convolutions, particularly when handling high-dimensional features. The ADown module significantly outperforms the downsampling convolution with a stride of 2 in terms of both parameter count and computational complexity, reducing the parameters and computational load while maintaining model performance. In the Backbone network of YOLOv8n, convolution layers (Conv) are replaced by ADown. ADown reduces computational complexity through efficient downsampling and convolution operations, enabling the model to execute more smoothly in real-time applications. By performing precise feature extraction, ADown ensures high accuracy in object detection, particularly in scenarios with dense products and complex backgrounds on goods shelf, where the model is able to consistently identify 109 different product categories.

### 2.2 Model compression for YOLOv8n-improved

The YOLOv8n-improved model built in our study can efficiently detect retail goods, but it still contains a large number of redundant parameters, making deployment difficult and real-time performance needs improvement. Considering the complex visual background in supermarkets and convenience stores and the limited resources of edge computing devices, this paper applies the channel pruning method from structured pruning [39] to compress the model, based on the YOLOv8 architecture. The principle of channel pruning is shown in Fig 7, which mainly includes three processes: sparse training, channel pruning and model fine-tuning.

To achieve rapid convergence during the sparsification training process, this paper introduces hyperparameters $\gamma$ and $\beta$ into the Batch Normalization (BN) layer of the YOLOv8n-improved network, and applies both scaling and shifting methods to normalize the channel data.

$$\boldsymbol{T}_{out} = \gamma \widehat{X} + \beta \tag{5}$$

$$\widehat{\boldsymbol{T}} = \frac{T_{in} - \mu_B}{\sqrt{\sigma_B^2 + \delta}} \tag{6}$$

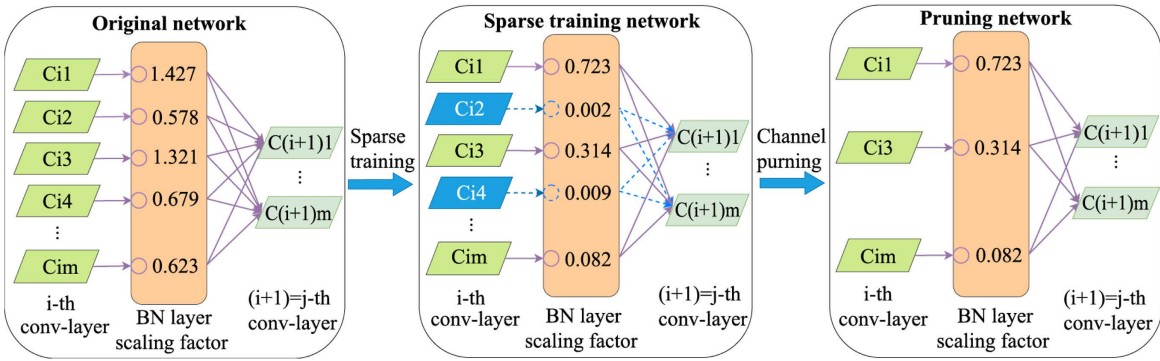

**Fig 7. Schematic of channel pruning.**

$$\mu_B = \frac{\sum_{i=1}^{n} x_i}{n} \tag{7}$$

$$\sigma_B = \frac{\sum_{i=1}^{n} (x_i - \mu_B)^2}{n} \tag{8}$$

where $T_{in}$ and $T_{out}$ denote the input and output of BN layer respectively, and $\hat{T}$ denotes The batch normalized value of the BN layer input.

The hyperparameter $\gamma$ is taken as the scaling factor. From equation (5), it can be observed that the output of the BN layer is positively correlated with $\gamma$. The smaller the value of $\gamma$, the closer the output approaches 0, indicating that the influence of the convolutional layer channel weights before the BN layer is reduced, allowing for their removal. Therefore, during the sparsification training process, the scaling factor $\gamma$ is selected as a proxy for channel selection [40]. An L1 regularization constraint on $\gamma$ is introduced into the loss function of the standard training, serving as a penalty term, and jointly training with the network weights to sparsify the scaling factor $\gamma$ [41]. The formula for the sparsification training loss function is as follows:

$$L = \sum_{(x,y)} (f(x,W),y)L' + \lambda \sum_{\gamma \in \Gamma} g(\gamma) \tag{9}$$

where $x$, $y$, $W$ and $f$ represent the input, target, weights, and output function of the original network, respectively. $\lambda$ is the sparsity factor, $g(\gamma)$ denotes the penalty function applied to the scaling factor $\gamma$, and $L'$ denotes the loss function of the original network. By multiple tests, $\lambda$ is set as an empirical value of 0.005, providing a reasonable trade-off between sparsity and the preservation of essential model information.

As shown in Fig 7, after sparse training, the scaling factor $\gamma$ of the BN layer approaches 0, indicating that the corresponding channels contribute little to the model. These channels, along with their corresponding inputs and outputs, can be pruned [42], resulting in a compressed model with lower complexity. Finally, the pruned model is fine-tuned to recover accuracy on retail products, thereby compressing the model, optimizing the structure, and improving real-time detection performance without affecting the detection accuracy.

## 3. Experiments

### 3.1 Locount datasets and experimental environment

In order to fully validate the performance of the proposed model in the retail scenario, Locount is selected as the experimental dataset. Locount is a specialized dataset designed for object detection and counting tasks in retail environments [21].

The Locount dataset comprises a diverse collection of real-world images captured in retail environments, featuring a wide range of objects such as products, shelves, and customers. After refinement, it includes 109 categories of retail products, as depicted in Fig 8. The dataset is designed to support model training for detecting and counting items in crowded and cluttered retail settings.

And annotations are provided in a TXT format, which facilitates training across various object detection models. This dataset is particularly valuable for tasks such as inventory management, product recognition, and customer behavior analysis. The selection of the Locount dataset is driven by the specific needs of retail applications, ensuring it provides a realistic and challenging environment for model evaluation. The dataset has been carefully divided into a training set, validation set, and test set, containing 3,500, 1,000, and 500 images, respectively. The detailed composition of the dataset, including the types of retail products and the complexity of the images, as well as the reasoning behind its selection over other retail datasets, highlights its suitability for addressing real-world challenges in retail environments.

To comprehensively validate the effectiveness of the proposed algorithm, the platforms and environmental parameters used during the training, testing, and model compression verification stages are detailed in Table 1. All parameter settings are standardized to ensure the validity and reliability of the experiments. This study sets random seed to the default fixed integer value of 0, which controls the reproducibility of the random processes and ensures that the experimental results can be consistently replicated.

### 3.2 Evaluation indicators

This study evaluates the performance of YOLOv8n and the improved model's accuracy using comprehensive metrics, including Precision, Recall, mean Average Precision (mAP) [43,44]. Precision is the proportion of true positive predictions

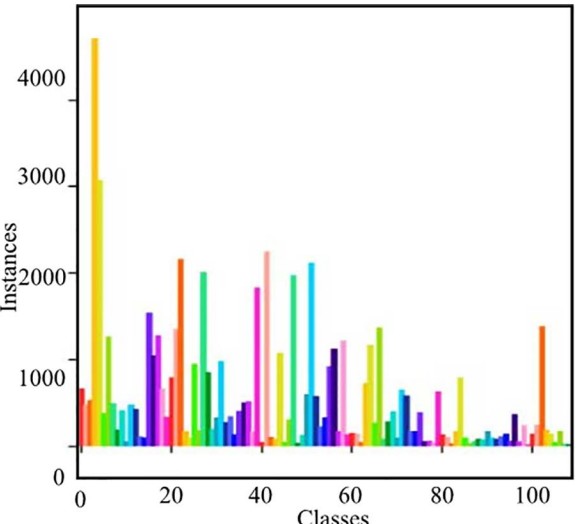

**Fig 8. The distribution of product categories and their quantities in the Locount dataset.**

**Table 1. Experimental configuration.**

| Category | Parameters | Result |
|---|---|---|
| Training parameters | Operating System | Ubuntu |
| | Optimizer | Stochastic gradient descent(SGD) |
| | Batch size | 16 |
| | Learning rate | 0.01 |
| | Epochs | 200 |
| | Input images size | 640×640 |
| | Momentum | 0.937 |
| Experimental environment parameters | GPU | NVIDIA GeForce RTX 3090 |
| | CPU | Intel(R) Xeon(R) Platinum 8350C |
| | CUDA | 11.7 |
| | cuDNN | 8.6 |
| | Pytorch | 2.0.0 |
| | Python | 3.8.17 |

out of all positive predictions made by the model, and Recall is the proportion of true positive predictions out of all actual positive instances in the dataset. mAP is used to measure the average precision of a model across different categories. mAP50 represents the average precision when the IoU threshold is set to 0.5, meaning that when IoU ≥ 0.5, the predicted bounding box is considered to be a match with the ground truth bounding box. A higher mAP value indicates that the model is able to maintain high precision at higher recall rates. On the other hand, mAP50-95 represents the average mAP value across different IoU thresholds, ranging from 0.5 to 0.95 with a step size of 0.05. The mathematical formulas for Precision, Recall, and mAP are as follows:

$$\mathrm{Pr}\,ecision = \frac{TP}{TP + FN} \tag{10}$$

$$\mathrm{Re}call = \frac{TP}{TP + FP} \tag{11}$$

$$mAP = \frac{1}{N}\int_0^1 \mathrm{Pr}\,ecision(\mathrm{Re}call)d(\mathrm{Re}call) \tag{12}$$

where TP, FN and FP denote the number of true positives, false negatives and false positives respectively; N is the number of object classes.

More importantly, Params (parameters) and GFLOPs are used as critical factors when evaluating the lightweight of models [45]. Params refers to the total number of parameters in the model, including weights and biases. GFLOPs refers to the number of billions of floating-point operations executed per second, which is used to measure the computational load during the model's forward propagation. Lightweight models require Params and GFLOPs to be as low as possible while maintaining performance. At the same time, real-time performance is also an important consideration for commodity targets detection tasks. Here, the number of images or data frames that the model can process per second, FPS (frames per second), is used as an indicator to evaluate the reasoning speed of the model.

## 3.3 Ablation experiments

During model training, the confidence threshold for the target object is set to the default value of 0.001, ensuring that small, difficult-to-detect targets are adequately considered as far as possible. The hyperparameters are shown in Table 1. The training follows Ultralytics' default of 100 epochs, ensuring optimal model performance with results that are consistently reproducible. The training curve of the improved model is shown in Fig 9.

An ablation study is conducted to evaluate and validate the effectiveness of the model lightweight improvements on baseline model. The experiment results on the Locount dataset reveal that each modification enhances the model's efficiency and inference speed, as is illustrated in Table 2. The baseline model (YOLOv8n) achieves a precision of 0.708, recall of 0.502, and an mAP50 of 0.600 with 3,343,451 parameters, 13.1 GFLOPs, and 110.4 fps. Incorporating DWConv results in a slight decrease in mAP50, but increases FPS to 123.4, while reducing parameters to 3,160,859. The addition of CSPHet and CBAM improves precision to 0.719 and reduces parameters further to 2,857,827, achieving a significant speed-up with 156.2 FPS, although it causes a slight decrease in mAP50 to 0.585. The final model, incorporating ADown, maintains competitive precision (0.71) and recall (0.502) while reducing parameters to 2,574,947 and GFLOPs to 8.1, resulting in an impressive increase in FPS to 277.7. This demonstrates the effectiveness of each modification in optimizing the model for faster inference with minimal sacrifice in detection performance.

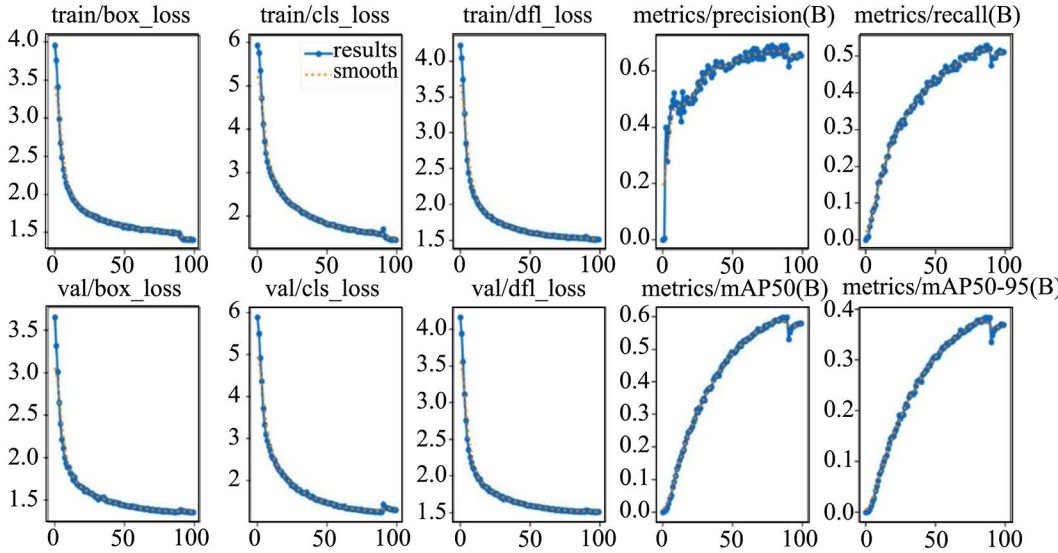

**Fig 9. The training curves of metrics in YOLOv8n-improved model.**

**Table 2. Ablation experiment results on Locount dataset.("D" denotes DWConv, "CC" denotes CSPHet-CBAM, and "A" denotes ADown).**

| Involved component (YOLOv8n-Baseline) | | | Metrics | | | | | | |
|---|---|---|---|---|---|---|---|---|---|
| +D | +CC | +A | Precision | Recall | mAP50 | mAP50-95 | Params | GFLOPs | FPS |
| ✗ | ✗ | ✗ | 0.708 | 0.502 | 0.600 | 0.395 | 3343451 | 13.1 | 110.4 |
| √ | ✗ | ✗ | 0.691 | 0.500 | 0.596 | 0.390 | 3160859 | 9.4 | 123.4 |
| √ | √ | ✗ | 0.719 | 0.499 | 0.585 | 0.373 | 2857827 | 8.8 | 156.2 |
| √ | √ | √ | 0.71 | 0.502 | 0.592 | 0.379 | **2574947** | **8.1** | **277.7** |

## 3.4 Model compression results

The pruning rate has varying degrees of impact on the detection performance of the model. Table 3 presents the results of pruning training on the YOLOv8n-improved model. As the pruning rate increases, the number of model parameters and floating-point operations (FLOPs) gradually decreases, leading to an improvement in detection speed. Notably, when the pruning rate is set below 0.5, the pruned model maintains a relatively stable mean Average Precision (mAP) of approximately 0.56, comparable to the original YOLOv8n-improved model. However, when the pruning rate exceeds this threshold, detection performance deteriorates sharply, with mAP dropping to 0.01. These results indicate that a pruning rate of 0.5 achieves an optimal balance, effectively reducing model complexity while minimizing accuracy loss in subsequent experiments. Fig 10 illustrates the changes in the number of channels after pruning, effectively reducing the total number of channels in the network from 8,432–5,260. This results in an average of 30 channels per layer, significantly optimizing the model's complexity.

Sparse training represents a trade-off between model accuracy and sparsity. Different sparsity strategies must be applied to various models in order to achieve a high degree of sparsity while maintaining strong accuracy for retail products. Generally, selecting a higher sparsity rate accelerates the sparsification process but also leads to a more significant decline in accuracy. Conversely, opting for a lower sparsity rate, though slower in achieving sparsity, results in a more gradual reduction in accuracy. Additionally, using a larger learning rate can expedite the sparsification process, while

**Table 3. Model parameters and performance metrics under different channel pruning rates.**

| Pruning rate | Result | | | | |
|---|---|---|---|---|---|
| | Parameter (Million) | GFLOPs | Model size(MB) | mAP50 | FPS |
| 0 | 2.57 | 8.1 | 5.18 | 0.592 | 277.7 |
| 0.1 | 2.42 | 7.7 | 5.06 | 0.588 | 282.9 |
| 0.2 | 2.32 | 7.2 | 4.72 | 0.582 | 295.2 |
| 0.3 | 2.29 | 6.9 | 4.4 | 0.580 | 296.8 |
| 0.4 | 2.18 | 6.7 | 4.31 | 0.575 | 321.5 |
| **0.5** | **2.11** | **6.6** | **4.24** | **0.563** | **351.2** |
| 0.6 | 1.98 | 4.3 | 3.75 | 0.330 | 340.5 |
| 0.7 | 1.70 | 3.8 | 3.1 | 0.120 | 332.1 |
| 0.8 | 1.22 | 3.2 | 2.2 | 0.010 | 335.8 |
| 0.9 | 1.16 | 2.9 | 0.98 | 0.010 | 317.0 |

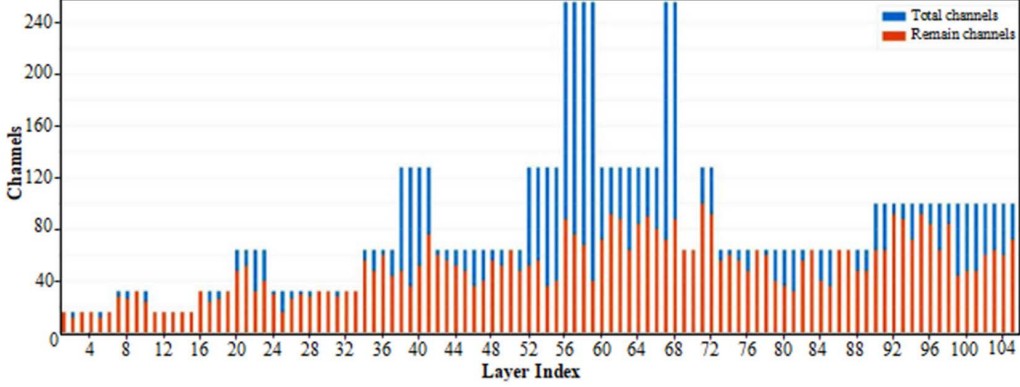

**Fig 10. Changes in the number of channels in the YOLOv8n-improved model at a pruning rate of 0.5.**

a smaller learning rate later in training helps to stabilize and recover accuracy. Through extensive experimentation, we ultimately adopt a constant-sparsity strategy for sparse training. This approach involves maintaining a fixed sparsity rate throughout the sparsification process, applying consistent additional gradients to the model. This results in a more uniform force distribution, leading to higher compression levels. The distribution and values of the Gamma coefficients for the BN layers prior to sparse training are illustrated in Fig 11.

Finally, this study performs fine-tuning on the pruned model for higher accuracy. Due to the loading of the pretrained weights after sparse training, the model's initial mAP is relatively high. After 200 epochs, mAP50 stabilizes at around 0.722 on test dataset of Locount, as is shown in Fig 12. After model compression, we get the final model, named LSR-YOLO (Lightweight Smart Retail YOLO model).

After model compression, the final model outperforms it in terms of precision (0.741), recall (0.649), and mAP50 (0.722), with a reduction in parameters to 2,114,768 and GFLOPs to 6.6, resulting in a significantly higher frame rate of 357.1 FPS than YOLOv8n-improved, as is shown in Table 4. The LSR-YOLOv8n model demonstrates superior detection performance, especially in recall and mAP50, while being more computationally efficient with fewer parameters and lower GFLOPs, making it more suitable for real-time shopping applications requiring both high accuracy and fast inference.

### 3.5 Comparative experiments

To further validate the effectiveness of the proposed LSR-YOLO method, experiments were conducted on the Locount dataset under the same experimental equipment and environment, comparing the proposed method with other mainstream object detection algorithms. The experimental results are shown in Table 5.

The LSR-YOLO model demonstrates significant advantages in terms of model compression and lightweight design when compared to other object detection algorithms. Specifically, LSR-YOLO, which is derived from the improved model after compression, offers notable improvements in various performance metrics while maintaining a reduced computational footprint. In terms of accuracy, LSR-YOLO achieves a Precision of 0.741 and a Recall of 0.649, surpassing other models such as YOLOv7 (0.738 and 0.584, respectively) and SR-YOLO (0.71 and 0.502, respectively). Furthermore, LSR-YOLO excels in mAP50 (0.722) and mAP50-95 (0.478), outperforming YOLOv8n-improved (0.592 and 0.379) and other models like YOLOv8n (0.600 and 0.395) in both metrics. This indicates that LSR-YOLO maintains high detection accuracy and robustness across a range of object sizes, even after compression.

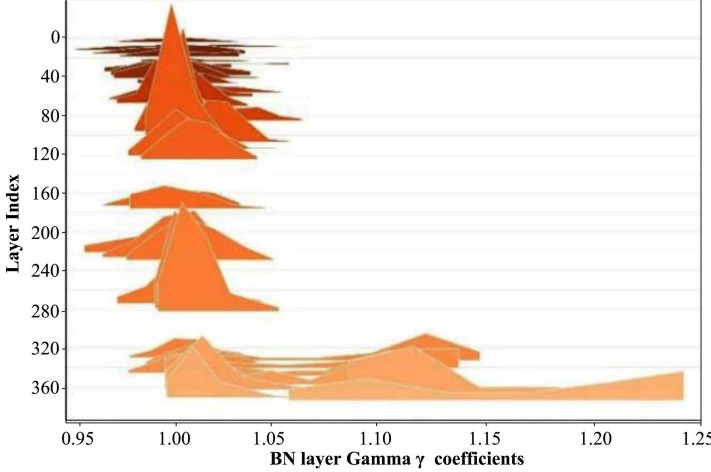

**Fig 11. The distribution of Gamma coefficients ($\gamma$) of the BN layers before YOLOv8n-improved model's sparse training.**

**Fig 12. Precision-Recall curve of LSR-YOLO model.**

**Table 4. Performance comparison of the improved model before and after pruning.**

| Model | Result | | | | | | |
|---|---|---|---|---|---|---|---|
| | Precision | Recall | mAP50 | mAP50-95 | Params | GFLOPs | FPS |
| YOLOv8n-improved | 0.71 | 0.502 | 0.592 | 0.379 | 2574947 | 8.1 | 277.7 |
| LSR-YOLOv8n (the final) | 0.741 | 0.649 | **0.722** | **0.478** | **2114768** | **6.6** | **357.1** |

**Table 5. Comparison of different object detection algorithms on Locount dataset.**

| Model | Result | | | | | | |
|---|---|---|---|---|---|---|---|
| | Precision | Recall | mAP50 | mAP50-95 | Params | GFLOPs | FPS |
| SSD 300 [16] | 0.623 | 0.435 | 0.548 | 0.366 | 4818100 | 35.8 | 121.6 |
| Faster-RCNN [24] | 0.613 | 0.422 | 0.538 | 0.351 | 41902000 | 267.3 | 87.7 |
| YOLOv5 [46,47] | 0.585 | 0.51 | 0.579 | 0.412 | **1906642** | 14.6 | 95.2 |
| YOLOv7 [48] | 0.738 | 0.584 | 0.581 | 0.429 | 37064324 | 10.5 | 119 |
| YOLOv8n [15] | 0.708 | 0.502 | 0.600 | 0.395 | 3343451 | 13.1 | 110.4 |
| Swin Transformer [49] | 0.772 | 0.376 | 0.571 | 0.337 | 45300000 | 213.0 | 72.9 |
| RT-DETR [50] | 0.753 | 0.361 | 0.563 | 0.447 | 32207735 | 103.9 | 28.4 |
| MobileNetV3 [51] | 0.462 | 0.498 | 0.588 | 0.423 | 10617000 | 166.0 | 57.8 |
| YOLO-NAS [52] | 0.146 | 0.792 | 0.572 | 0.398 | 19066967 | 17.2 | 29.4 |
| YOLOv11 [53] | 0.552 | 0.538 | 0.58 | 0.411 | 2650315 | 6.7 | 340.3 |
| **YOLOv8n-improved** | 0.71 | 0.502 | 0.592 | 0.379 | 2574947 | 8.1 | 277.7 |
| **LSR-YOLO** | **0.741** | **0.649** | **0.722** | **0.478** | 2114768 | **6.6** | **357.1** |

It is worth noting that transformer-based models face the issue of excessive computational complexity, which is likely attributed to their too deep network layers. Meanwhile, existing lightweight models such as YOLO-NAS, despite having fewer parameters, suffer from slow inference speeds, rendering them impractical for real-time retail applications.

One of the key advantages of LSR-YOLO lies in its model size and computational efficiency. With only 2,114,768 parameters and 6.6 GFLOPs, LSR-YOLO is significantly more lightweight compared to models like Faster-RCNN (41,902,000 parameters and 267.3 GFLOPs) and YOLOv7 (37,064,324 parameters and 10.5 GFLOPs). This reduction in parameters and computational complexity not only enhances the speed of the model but also makes it suitable for deployment in resource-constrained environments.

In terms of FPS, LSR-YOLO achieves a remarkable 357.1 fps, which is significantly higher than SR-YOLO (277.7 fps), YOLOv7 (119 fps), and other models such as YOLOv5 (95.2 fps). This demonstrates that LSR-YOLO can process images at a much faster rate, enabling real-time detection even in scenarios with limited computational resources.

Analyzing the GFLOPs and FPS metrics in Table 5, the YOLO series models exhibit significantly superior performance compared to SSD300 and Faster R-CNN. However, YOLOv5 demonstrates a lower FPS compared to other YOLO models. Modern models such as YOLOv8n and YOLOv11 face challenges in balancing computational complexity with inference speed, which often becomes a bottleneck in practical retail applications. For instance, YOLOv8n achieves lower computational complexity but at the cost of reduced accuracy, whereas YOLOv8x delivers higher accuracy yet incurs significantly greater computational demands, leading to slower inference.

LSR-YOLO addresses these shortcomings through lightweight architectural enhancements. The integration of DWConv in the Neck, which decouples convolution operations across spatial and channel dimensions, substantially reduces parameter count and computational cost without compromising accuracy. To preserve detection precision, the CSPHet-CBAM attention mechanism is incorporated, ensuring robust feature representation despite architectural simplifications. Moreover, the application of a channel pruning algorithm removes redundant channels, further reducing computation and improving real-time performance—an essential requirement for retail environments where speed is critical.

In addition to the overall performance comparison, a subsequent visual analysis was conducted, as shown in Fig 13. Considering the requirements of practical applications, this study selects some models with significant value for mobile robots or devices based on their performance metrics. Notably, when handling blurred images and densely arranged products on shelves, models from YOLOv7 to YOLOv11 performed worse than the final model, exhibiting a high number of missed detections. Moreover, YOLOv8n and YOLOv11 show weaker performance in detecting objects at the image edges. In contrast, the pruned YOLOv8n-improved model effectively mitigated this issue. Regarding occlusion detection, the LSR-YOLO model demonstrated a stronger focus mechanism, enabling robust detection of occluded objects.

### 3.6 Generalization validation

Considering the generalization ability of the model and its scalability to accommodate the continuously growing variety of products, the COCO 2017 dataset [22] is used to validate the generalization performance of the LSR-YOLO method in our study. The COCO dataset is one of the most widely used and comprehensive datasets for object detection tasks in computer vision, specifically designed to provide high-quality annotations for a wide variety of real-world scenes. And COCO includes over 330,000 images, of which more than 200,000 are labeled. Furthermore, contains over 80 object categories, which include everyday objects such as people, animals, vehicles, and household items.

The training configuration is summarized in Table 1. The model is trained for 200 epochs using the SGD optimizer with a batch size of 16 and an initial learning rate of 0.01. Input images are resized to the default resolution of 640×640 pixels, while a momentum of 0.937 is applied to accelerate convergence.

**3.6.1 Ablation experiments on COCO.** To validate the performance gains brought by the optimization strategies, including the lightweight convolution DWConv, the newly designed CSPHet-CBAM, and the ADown downsampling

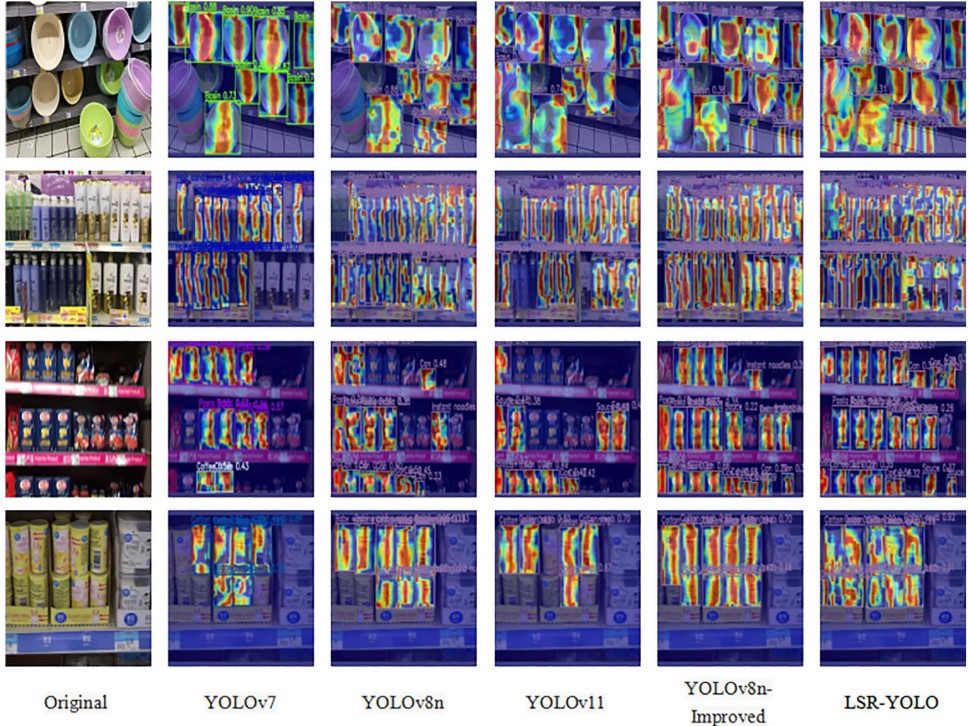

| Original | YOLOv7 | YOLOv8n | YOLOv11 | YOLOv8n-Improved | LSR-YOLO |

**Fig 13. Comparison of heatmaps on retail products for different object detection models.**

mechanism, ablation experiments were conducted on the COCO dataset. The experimental results are presented in Table 6. These strategies aim to enhance the model's performance by improving efficiency and accuracy.

By introducing the DWConv module (YOLOv8n+DWConv), precision decreases slightly to 0.586, but the model also benefits from a reduction in parameters to 2,969,312 and GFLOPs to 8.5, while the FPS increases to 200. These results demonstrate that the DWConv module contributes to a more efficient model, improving recall and computational performance without significantly sacrificing precision. Adding the CSPHet-CBAM module results in further reductions in parameters (2,666,280) and GFLOPs (7.9), as well as a significant increase in FPS to 312.5. Although precision slightly decreases to 0.592 and recall remains relatively stable at 0.45, the model demonstrates a notable reduction in computational complexity and improved inference speed, while mAP50 stays competitive at 0.487. The decrease in mAP50-95 to 0.326 may be a trade-off due to the increased focus on efficiency. Finally, the inclusion of the ADown downsampling mechanism leads to a further reduction in parameters (2,383,400) and GFLOPs (7.2), with the highest FPS of 322.5. While precision drops to 0.577 and recall to 0.438, the model retains an acceptable mAP50 of 0.490 and shows a slight

**Table 6. Ablation experiment results on COCO2017 dataset.("D" denotes DWConv, "CC" denotes CSPHet-CBAM, and "A" denotes ADown).**

| Involved component (YOLOv8n-Baseline) | | | Metrics | | | | | | |
|---|---|---|---|---|---|---|---|---|---|
| +D | +CC | +A | Precision | Recall | mAP50 | mAP50-95 | Params | GFLOPs | FPS |
| ✗ | ✗ | ✗ | 0.604 | 0.446 | 0.493 | 0.342 | 3151904 | 9.6 | 112.3 |
| √ | ✗ | ✗ | 0.586 | 0.457 | 0.491 | 0.338 | 2969312 | 8.5 | 200 |
| √ | √ | ✗ | 0.592 | 0.450 | 0.487 | 0.326 | 2666280 | 7.9 | 312.5 |
| √ | √ | √ | 0.577 | 0.438 | 0.490 | 0.335 | 2383400 | 7.2 | 322.5 |

improvement in mAP50-95 to 0.335. These findings illustrate the effectiveness of the ablation study in balancing performance and efficiency.

**3.6.2 Comparative experiments on COCO.** The comparison between LSR-YOLO and other models on the COCO2017 dataset highlights the significant lightweight advantages and improved generalization performance of the proposed LSR-YOLO model, especially in terms of model compression, as is illustrated in Table 7. By model compression, it results in a substantial reduction in both parameters and computational complexity while maintaining competitive detection performance. Specifically, LSR-YOLO achieves a precision of 0.564, recall of 0.439, and mAP50 of 0.495, with only 1,768,097 parameters and 5.1 GFLOPs, while maintaining a frame rate of 333.3 FPS. This showcases its efficiency in terms of both accuracy and computational cost.

In comparison to other object detection models, LSR-YOLO significantly reduces the number of parameters and GFLOPs while retaining a high mAP50, especially when compared to models like SSD and Faster-RCNN, which have much larger parameter counts (34,305,000 and 41,753,000, respectively). Additionally, LSR-YOLO delivers faster inference with a higher FPS than most models, including YOLOv5 and YOLOv7, which are known for their lightweight architecture. Meanwhile, transformer-based models still encounter challenges stemming from their excessive computational demands and limited inference speed (low FPS).

Notably, while LSR-YOLO sacrifices a slight reduction in precision (0.564) compared to YOLOv5 (0.583) and YOLOv7 (0.570), it compensates for this by offering a substantial reduction in computational complexity (in terms of both parameters and GFLOPs), making it highly suitable for deployment in resource-constrained environments or applications where speed is critical.

Finally, a visualization analysis was conducted in the COCO dataset, as shown in Fig 14, with a particular focus on indoor environments to better simulate real-world retail settings. Given that SSD 300 and Faster R-CNN exhibited suboptimal performance in terms of the evaluation metrics presented in Table 7, and that YOLOv5 has a significantly larger number of parameters compared to the improved model, these models were excluded from the visualization heatmap analysis. YOLOv7 and its subsequent models perform similarly, though it is noted that YOLOv7 and YOLOv11 experience a small number of false positives (as seen in the floor area in the second subplot). Models that are solely optimized for lightweight network improvements also show this behavior. Additionally, YOLOv11 exhibits a few missed detections, as evidenced by its failure to detect the book on the left side of the table in the third subplot. It is evident that the final model obtained through model compression demonstrates clear advantages in performance, achieving more stable detection accuracy with a lighter network structure and faster detection speed.

**Table 7. Comparison of detection performance between LSR-YOLO and other models on COCO2017 dataset.**

| Model | Result | | | | | | |
|---|---|---|---|---|---|---|---|
| | Precision | Recall | mAP50 | mAP50-95 | Params | GFLOPs | FPS |
| SSD [16] | 0.632 | 0.341 | 0.398 | 0.217 | 34305000 | 34.3 | 110.1 |
| Faster-RCNN [24] | 0.660 | 0.383 | 0.486 | 0.325 | 41753000 | 187 | 31.8 |
| YOLOv5 [46,47] | 0.583 | 0.442 | 0.487 | 0.329 | 7225885 | 16.4 | 289.4 |
| YOLOv7 [48] | 0.570 | 0.482 | 0.483 | 0.324 | 6221370 | 13.7 | 185.1 |
| YOLOv8n[15] | 0.604 | 0.446 | 0.493 | **0.342** | 3151904 | 9.6 | 112.3 |
| Swin Transformer [49] | 0.365 | 0.420 | 0.447 | 0.254 | 45300000 | 194.0 | 10.2 |
| RT-DETR [50] | 0.794 | 0.144 | 0.470 | 0.415 | 32148140 | 103.8 | 91.7 |
| MobileNetV3 [51] | 0.398 | 0.406 | 0.441 | 0.289 | 10016000 | 136.0 | 26.7 |
| YOLO-NAS [52] | 0.034 | 0.756 | 0.446 | 0.305 | 19053888 | 17.2 | 45.8 |
| YOLOv11 [53] | 0.601 | 0.442 | 0.484 | **0.346** | 2616248 | 6.5 | 318.7 |
| **YOLOv8n-improved** | 0.577 | 0.438 | 0.490 | 0.335 | 2383400 | 7.2 | 322.5 |
| **LSR-YOLO** | 0.564 | 0.439 | **0.495** | 0.341 | **1768097** | **5.1** | **333.3** |

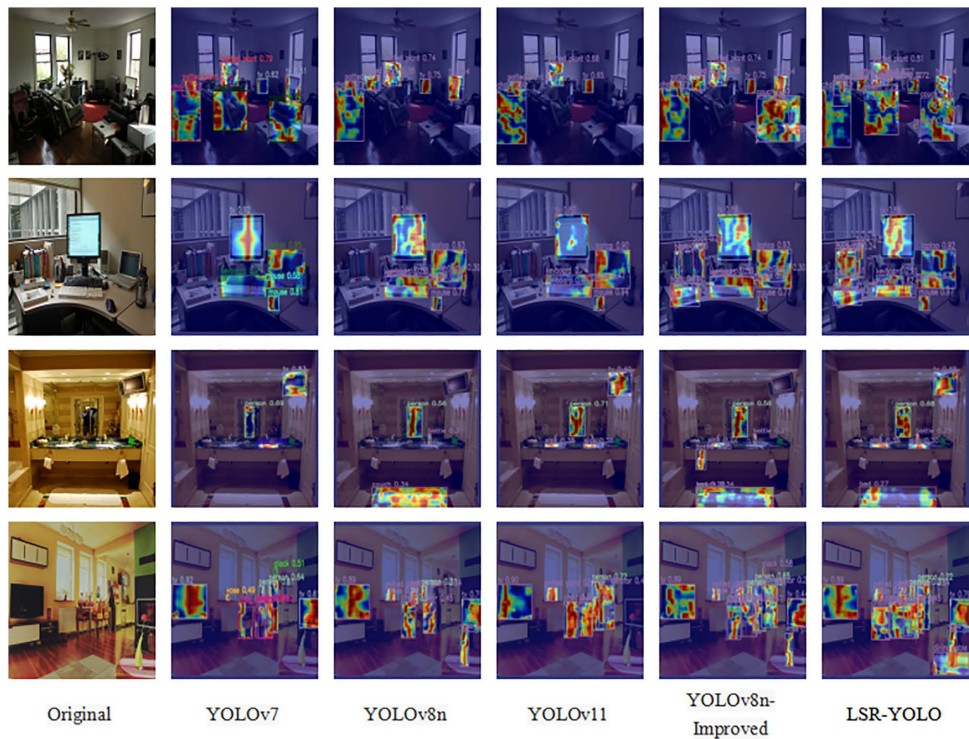

| Original | YOLOv7 | YOLOv8n | YOLOv11 | YOLOv8n-Improved | LSR-YOLO |

**Fig 14. Comparison of heatmaps for different object detection models on COCO scenes.**

## 3.7 The 5-fold cross validation

For the training and validation sets, comprising a total of 4,500 images, 5-fold cross-validation is employed to ensure more reliable performance evaluation. The final cross-validation results on the Locount dataset are summarized in Table 8.

The precision, recall, mAP50, and mAP50-95 values exhibit only minor variations between the original and cross-validation results, reinforcing the stability of the LSR-YOLO model. Specifically, precision improves slightly ($0.741 \rightarrow 0.749$) and mAP50-95 increases ($0.478 \rightarrow 0.493$), whereas recall ($0.649 \rightarrow 0.609$) and mAP50 ($0.722 \rightarrow 0.680$) show small decreases. However, these fluctuations are negligible and do not compromise the model's overall consistency. Moreover, the minimal differences observed in parameters, CFLOPs, and FPS further confirm that LSR-YOLO maintains robust performance across different data splits.

## 3.8 Potential trade-offs and limitations

By balancing the trade-off between common scenario coverage and lightweight design, the LSR-YOLO model demonstrates strong performance in standard visual environments and maintains reasonable accuracy under motion blur and dense object arrangements. However, its detection capability is comparatively weaker in scenarios involving object stacking, occlusion, or low-light conditions.

**Table 8. The 5-fold cross validation result of LSR-YOLO on Locount dataset.**

| Model | Precision | Recall | mAP50 | mAP50-95 | Parameters | CFLOPs | FPS |
|-------|-----------|--------|-------|----------|------------|--------|-----|
| LSR-YOLO | 0.741 | 0.649 | 0.722 | 0.478 | 2114768 | 6.6 | 357.1 |
| **5-Fold CV** | 0.749 | 0.609 | 0.680 | 0.493 | 2485861 | 7.8 | 349.7 |

As shown in the Fig 15, in the case of multiple bowls stacked and occluded, partial missed detections occur, and the bowls in the rear rows are not detected.

In low-light conditions, as illustrated in Fig 16, the missed detection problem becomes more pronounced in LSR-YOLO. Under insufficient lighting, the model's ability to capture product color features diminishes, and confidence scores are adversely affected.

In the next phase of this research, a targeted feature sensitivity analysis will be performed on occluded object samples, followed by architectural enhancements to the model. In addition, corresponding training samples will be incorporated into the dataset to improve robustness. For products in low-light environments, light enhancement algorithms may be explored, and multimodal approaches integrating infrared and visible-light imaging could also be considered.

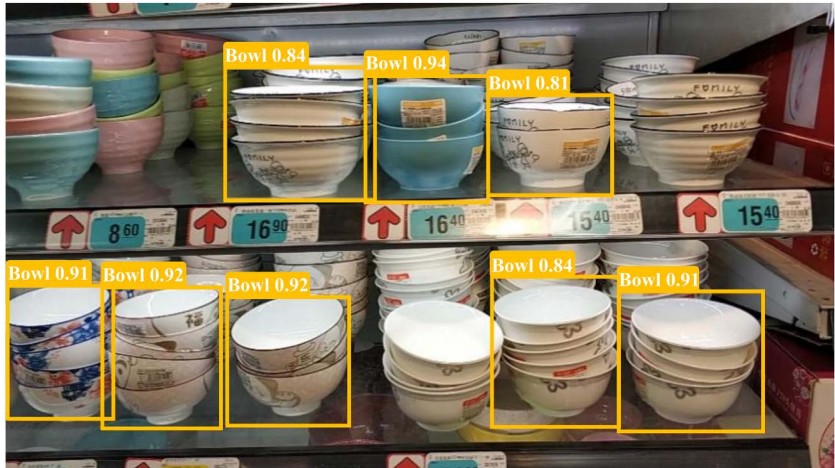

**Fig 15. Examples of negative samples for product detection with multiple object occlusions.**

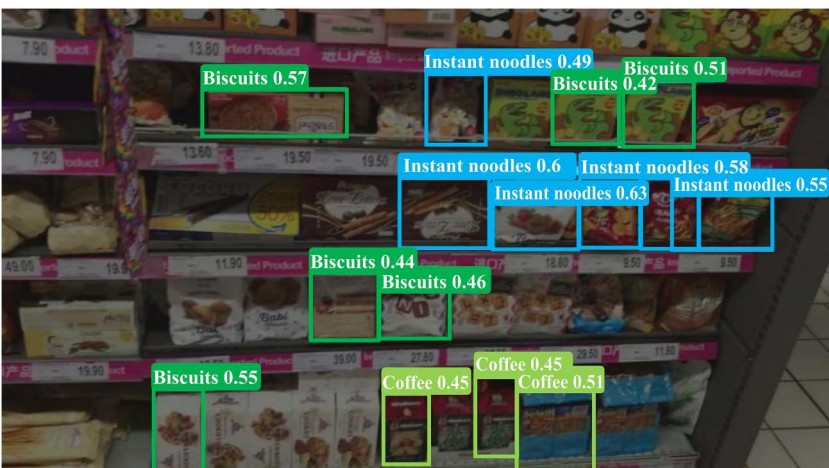

**Fig 16. Examples of negative samples for product detection under low-light conditions.**

### 3.9 Preliminary deployment on edge computing device

To evaluate the feasibility of deploying LSR-YOLO in robotics and edge computing applications, this study utilized the low-power and cost-effective RK3568 device for preliminary testing. The RK3568 is equipped with a quad-core 64-bit Cortex-A55 processor (2.0 GHz) and an integrated Rockchip NPU capable of 1 TOPS. For deployment, the LSR-YOLO model was first converted from the.pt format to.onnx before inference testing. The experimental results show that the model comprises 2,113,616 parameters and 3.80 GLOPs, achieving an inference speed of 10.6 FPS. Despite the RK3568 being a relatively low-performance edge device, these results demonstrate that LSR-YOLO meets the practical requirements for robotic inspection and handheld imaging applications, with higher-performance hardware expected to further enhance performance.

## 4. Conclusions

To address challenges in retail product recognition, this study proposes LSR-YOLO (Lightweight Smart Retail YOLO), a novel object detection method specifically designed for efficient retail product detection based on the YOLOv8n model. First, lightweight architectural enhancements are introduced to optimize computational performance without compromising accuracy. Depthwise Convolution (DWConv) layers at multiple scales are incorporated into YOLOv8n's Neck, decoupling spatial and channel convolutions, thereby reducing parameters and computational overhead. To mitigate potential accuracy loss from neck lightweighting, an attention mechanism—CSPHet-CBAM—is integrated to enhance feature representation. In addition, the bulky backbone network is simplified using ADown, a downsampling optimization that combines average pooling and max pooling. Finally, a channel pruning algorithm with a pruning rate of 50% is applied to the improved YOLOv8n model, effectively identifying and removing less informative channels. This ensures that the pruned model sustains high accuracy while substantially enhancing real-time detection performance.

Extensive experiments on the Locount dataset demonstrate that the proposed LSR-YOLO achieves a substantial improvement in inference speed, reaching 357.1 FPS—an increase of 246.7 FPS over the YOLOv8n baseline (110.4 FPS)—while simultaneously enhancing accuracy. Its lightweight design is further evidenced by 2,114,768 parameters and 6.6 GFLOPs, making it significantly more efficient than advanced models such as YOLOv11. The model's generalization ability was further validated on the widely used COCO 2017 dataset, where LSR-YOLO consistently outperformed state-of-the-art models in both accuracy and efficiency. Specifically, it achieved a mAP50 of 0.495 with only 1,768,097 parameters and 5.1 GFLOPs, while sustaining an inference speed of 333.3 FPS. These results highlight the potential of LSR-YOLO as an effective solution for real-time object detection in autonomous retail robotics.

Future research on object detection in retail environments offers several promising directions. A key priority is enhancing the generalization ability of the LSR-YOLO model to better adapt to increasingly diverse retail scenarios, including varying lighting conditions and a wider range of product categories. Another potential direction involves exploring hybrid approaches that integrate LSR-YOLO with state-of-the-art methods, such as transformer-based architectures, to improve feature extraction and robustness. Furthermore, embedding the model into embodied intelligence systems could provide richer visual perception in shopping scenarios, enabling more advanced applications in autonomous retail robotics.

### Acknowledgments

The authors would like to express their sincere gratitude to the Centre of Excellence for Research, Value Innovation and Entrepreneurship (CERVIE), UCSI University, Malaysia, for supporting this project under the code REIG-FETBE-2025/015.

### Author contributions

**Conceptualization:** Yawen Zhao, Mahmud Iwan Solihin, Defu Yang, Anton Satria Prabuwono.

**Data curation:** Yawen Zhao.

**Formal analysis:** Yawen Zhao, Mahmud Iwan Solihin.

**Funding acquisition:** Yawen Zhao, Mahmud Iwan Solihin, Dini Handayani.

**Investigation:** Yawen Zhao.

**Methodology:** Yawen Zhao, Mahmud Iwan Solihin, Defu Yang.

**Project administration:** Dini Handayani.

**Resources:** Yawen Zhao, Bingyu Cai.

**Software:** Yawen Zhao, Bingyu Cai.

**Supervision:** Mahmud Iwan Solihin, Chow Li Sze, Anton Satria Prabuwono.

**Validation:** Defu Yang.

**Visualization:** Yawen Zhao, Defu Yang.

**Writing – original draft:** Yawen Zhao, Defu Yang.

**Writing – review & editing:** Mahmud Iwan Solihin, Chow Li Sze, Dini Handayani, Anton Satria Prabuwono.

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
