## [Decision Letter · Decision Letter 0]

14 Jun 2025

Dear Dr. Solihin,

Thank you for submitting your manuscript to PLOS ONE. After careful consideration, we feel that it has merit but does not fully meet PLOS ONE’s publication criteria as it currently stands. Therefore, we invite you to submit a revised version of the manuscript that addresses the points raised during the review process.

The reviewers raised comments that need to be addressed.

We look forward to receiving your revised manuscript.

Kind regards,

Alberto Marchisio

Academic Editor

PLOS ONE

Journal Requirements:

Reviewers' comments:

Reviewer's Responses to Questions

**Comments to the Author**

1. Is the manuscript technically sound, and do the data support the conclusions?

Reviewer #1: Yes

Reviewer #2: Yes

Reviewer #3: Yes

Reviewer #4: Yes

Reviewer #5: Partly

2. Has the statistical analysis been performed appropriately and rigorously?

Reviewer #1: Yes

Reviewer #2: Yes

Reviewer #3: Yes

Reviewer #4: Yes

Reviewer #5: No

3. Have the authors made all data underlying the findings in their manuscript fully available?

Reviewer #1: Yes

Reviewer #2: Yes

Reviewer #3: No

Reviewer #4: Yes

Reviewer #5: Yes

4. Is the manuscript presented in an intelligible fashion and written in standard English?

Reviewer #1: Yes

Reviewer #2: Yes

Reviewer #3: Yes

Reviewer #4: Yes

Reviewer #5: Yes

Reviewer #1: The article proposes LSR-YOLO, a lightweight and fast object detection model based on YOLOv8n, optimized for retail product detection using techniques like DWConv, CSPHet-CBAM, ADown, and channel pruning, achieving a high inference speed of 357.1 FPS with a mAP50 of 72.2% on the Locount dataset. It demonstrates superior efficiency and accuracy compared to other models like YOLOv11, with reduced parameters (2,114,768) and GFLOPs (6.6), validated further on the COCO dataset for generalization.

Here are some major comments:

- Figure 2 contains a typographical error where "Contact" is used instead of "Concat" to describe the concatenation operation in the improved YOLOv8n model architecture. Correcting this error and ensuring consistent terminology across all figures and text would enhance the article’s professionalism and clarity.

- The article could improve clarity by providing more detailed explanations of technical components like CSPHet-CBAM and ADown. For instance, a brief description of how CSPHet integrates with CBAM or how ADown's hybrid pooling strategy specifically enhances feature extraction would make the methodology more accessible to readers unfamiliar with these concepts.

- The Locount dataset is central to the experiments, but its description is brief. Providing more details about its composition (e.g., types of retail products, image complexity, or annotation specifics) and why it was chosen over other retail datasets would strengthen the justification for its use and enhance reproducibility.

- The article mentions trade-offs in accuracy due to model compression but does not discuss potential limitations or failure cases of LSR-YOLO, such as performance in extreme lighting conditions or with highly occluded objects. Including a section on limitations and challenges would provide a more balanced perspective.

- While Table 4 compares LSR-YOLO with other models, the discussion could be expanded to explain why modern models (e.g., Yolov8n, Yolov11n) perform poorly or how LSR-YOLO’s improvements specifically address their shortcomings. This would provide deeper insights into the proposed model’s advantages.

- The article should provide a detailed explanation for the significantly lower performance of YOLOv8n and YOLOv11 on the COCO2017 dataset compared to their original reported results, potentially addressing factors such as differences in training configurations, dataset preprocessing, or evaluation metrics to ensure transparency and contextualize the findings.

- The article emphasizes deployment on edge devices but lacks details on real-world implementation challenges, such as hardware constraints or integration with retail robotic systems. Adding a brief discussion on practical deployment considerations would enhance the article’s applicability.

Reviewer #2: I must say that the revisions made to the article have been executed exceptionally well. The changes have addressed all the concerns and suggestions that were previously raised, resulting in a much more polished and coherent piece of work. As I have thoroughly reviewed the revised version, I am satisfied with the improvements and clarity it now possesses. Therefore, I have no further questions or additional points of clarification regarding this article.

Reviewer #3: This manuscript presents LSR-YOLO, an improved and lightweight version of YOLOv8n tailored for real-time retail product detection. The model integrates architectural modifications including DWConv, CSPHet-CBAM attention, and ADown pooling for efficiency. The authors further apply structured channel pruning to compress the model, achieving significant gains in speed and parameter reduction without compromising detection accuracy. Extensive evaluation on the Locount and COCO datasets supports the claims.

follwoing comments may be addressed before publication at plosOne

1. The individual novelty of components (DWConv, CBAM, ADown) is limited as each is previously known; the contribution lies in their integration and optimization for retail.

2. Comparison with other lightweight models (e.g., MobileNetV3, EfficientDet-Lite, YOLO-NAS) is missing and could contextualize performance better.

3. authors didnot performed confidence intervals or statistical significance testing across multiple runs.

4. please add training curves to give better look

5. There are occasional grammatical errors. Do a thorugh proof read before revision submission. For example:

“...processing at 357.1 fps compared to YOLOv8n baseline’s 78.8 fps...” → should be 78.7 fps for consistency.

you should use consistent notation: “YOLOv8n-improved” vs. “improved YOLOv8n”.

Reviewer #4: The manuscript introduces LSR-YOLO, a lightweight and efficient object detection model tailored for real-time retail applications. The model builds upon YOLOv8n and incorporates three key enhancements—DWConv, CSPHet-CBAM attention module, and ADown for downsampling—followed by structured channel pruning. The proposed approach is compelling and well-validated through multiple experiments.

Strengths:

The manuscript addresses a relevant and practical challenge in real-time retail object detection.

Detailed methodology with clear architectural diagrams (DWConv, CBAM, ADown).

Strong empirical evaluation using appropriate metrics and datasets (Locount and COCO).

Clear improvement over baseline models in terms of both speed and accuracy.

Thoughtful ablation and pruning analysis enhance the credibility of the findings.

Suggestions for Improvement:

Language: Numerous grammatical errors and stylistic inconsistencies must be corrected. Consider professional editing.

Clarity: The description of some modules (e.g., CSPHet-CBAM) is dense; more intuitive explanations or simplified illustrations could help.

References: Many citations are to arXiv or conference proceedings without DOI or full publication details. Ensure formatting aligns with journal standards.

Figures: Figures mentioned (e.g., Figure 1 to Figure 12) should be made visible in the manuscript file to fully evaluate their clarity and contribution.

Model limitations: Discuss potential trade-offs or limitations, such as potential accuracy drops under severe occlusion or small object detection.

Reviewer #5: 1. Limited novelty in methodology: While CSPHet-CBAM is presented as novel, it is primarily a hybrid of known techniques (CBAM + HetConv).

Recommend the authors clarify how this differs from past works that use similar hybrid attention or mixed convolution strategies.

2. Overemphasis on FPS: The manuscript prioritizes FPS gains, but real-world inference latency, energy consumption, or deployment on actual hardware (e.g., Jetson Nano, Raspberry Pi) would strengthen the practical relevance.

3. Pruning Analysis Needs Depth: Sparse training is covered, but more discussion on pruning criteria (e.g., L1 norm threshold choice) and stability across runs would improve robustness claims.

4. Missing Comparisons with Transformer-based Detectors: While YOLO-family comparisons are extensive, the paper does not consider newer lightweight transformer-based models (e.g., YOLOS, Mobile-DETR). A discussion of these would be valuable.

5. No Variance or Confidence Reporting: The results are reported as point estimates (e.g., “Precision = 0.741”), but no standard deviation, confidence interval, or multiple-run averages are reported.

This is a concern, especially since deep learning training can produce different results depending on initialization or data shuffling.

6. No Statistical Tests: There is no use of statistical hypothesis testing (e.g., t-tests, ANOVA) to determine if improvements over baseline models are statistically significant.

While not always required in ML papers, it would strengthen the claim that LSR-YOLO consistently outperforms others.

7. No Cross-Validation: The model is evaluated using a fixed train-validation-test split. Cross-validation or even repeating experiments with different splits would give more reliable performance estimates.

8. Missing Random Seed or Reproducibility Details: There is no mention of whether a fixed random seed was used, nor is there discussion of run-to-run variability. Reproducibility is a key part of rigor in ML research.

9. Grammatical Errors and Awkward Phrasing: Frequent issues with subject-verb agreement, article usage, and sentence structure.

Example:

"We uses DWConv as convolution layers..." → should be "We use DWConv as convolutional layers..."

"With only 2,114,768 parameters and 6.6 GFLOPs, LSR-YOLO is significantly more lightweight compared to advanced models including YOLOv11."

→ "…significantly lighter than advanced models such as YOLOv11."

10. Redundancy and Wordiness: Some paragraphs are overly long and repeat information (e.g., repeating FPS comparisons in several sections).

Sentences like "This design enhances the efficiency of the model without sacrificing performance" are generic and overused.

11. Inconsistent Terminology: Some abbreviations (e.g., “YOLOv8n-improved,” “SR-YOLO,” “LSR YOLO”) are used inconsistently or introduced without clear distinction.

**Do you want your identity to be public for this peer review?** For information about this choice, including consent withdrawal, please see our Privacy Policy

Reviewer #1: **Yes: ** Hoang Van Thanh

Reviewer #2: No

Reviewer #3: **Yes: ** Dr. Nadeem Iqbal Kajla

Reviewer #4: No

Reviewer #5: No

---

## [Author Response · Author response to Decision Letter 1]

20 Aug 2025

The response to reviewers is attached here.

---

## [Decision Letter · Decision Letter 1]

25 Sep 2025

LSR-YOLO: a lightweight and fast model for retail products detection

PONE-D-25-15496R1

Dear Dr. Solihin,

We’re pleased to inform you that your manuscript has been judged scientifically suitable for publication and will be formally accepted for publication once it meets all outstanding technical requirements.

Kind regards,

Joanna Tindall, PhD

Staff Editor

PLOS ONE

Additional Editor Comments (optional):

Reviewer #1:

Reviewer #5:

Reviewers' comments:

Reviewer's Responses to Questions

**Comments to the Author**

Reviewer #1: All comments have been addressed

Reviewer #5: All comments have been addressed

2. Is the manuscript technically sound, and do the data support the conclusions?

Reviewer #1: Yes

Reviewer #5: Yes

3. Has the statistical analysis been performed appropriately and rigorously?

Reviewer #1: Yes

Reviewer #5: Yes

4. Have the authors made all data underlying the findings in their manuscript fully available?

Reviewer #1: Yes

Reviewer #5: Yes

5. Is the manuscript presented in an intelligible fashion and written in standard English?

Reviewer #1: Yes

Reviewer #5: Yes

Reviewer #1: The authors have thoroughly addressed all of the reviewers’ concerns. I have no further suggestions for improvement. The manuscript is now suitable for publication.

Reviewer #5: The revised text is clearer, with structured sections, figures, equations, and tables. Some minor grammatical issues remain (e.g., “a attention mechanism” should be “an attention mechanism”), but overall readability is much improved.

**Do you want your identity to be public for this peer review?** For information about this choice, including consent withdrawal, please see our Privacy Policy

Reviewer #1: No

Reviewer #5: No

---

## [Editor Report · Acceptance letter]

PONE-D-25-15496R1

PLOS ONE

Dear Dr. Solihin,

I'm pleased to inform you that your manuscript has been deemed suitable for publication in PLOS ONE. Congratulations! Your manuscript is now being handed over to our production team.

Kind regards,

on behalf of

Dr Joanna Tindall

Staff Editor

PLOS ONE